# Factors Related to Anxiety in Paediatric Patients and Their Parents before and after a Modified Ravitch Procedure—A Single-Centre Cohort Study

**DOI:** 10.3390/ijerph192416701

**Published:** 2022-12-12

**Authors:** Dariusz Fenikowski, Lucyna Tomaszek

**Affiliations:** 1Department of Thoracic Surgery, Institute of Tuberculosis and Lung Diseases, Rabka-Zdrój Branch, 34-700 Rabka-Zdrój, Poland; 2Faculty of Medicine and Health Sciences, Andrzej Frycz Modrzewski Krakow University, 30-705 Kraków, Poland

**Keywords:** perioperative anxiety, factors, parents, children, patients’ satisfaction, Ravitch procedure

## Abstract

Objective. To assess the factors related to perioperative state anxiety in paediatric patients and their parents. Methods. A cohort study was conducted on paediatric patients 9–17 years of age, who underwent the modified Ravitch procedure (*n* = 96), and their parents (*n* = 96). The level of anxiety was measured using the State-Trait Anxiety Inventory questionnaire. Multivariable linear regression models were calculated to find the relationships between the pre- and postoperative state anxiety of the patients/parents and the independent variables, both demographic (age, gender) and clinical (e.g., postoperative pain, trait anxiety). Results. Preoperative anxiety in the paediatric patients was positively correlated with their trait anxiety (β = 0.47; 95% CI: 0.29 to 0.64) and preoperative parental anxiety (β = 0.24; 95% CI: 0.07 to 0.42). The high level of preoperative anxiety (vs. low and moderate) (β = 0.40; 95% CI: 0.22 to 0.58), trait anxiety (β = 0.22; 95% CI: 0.04 to 0.40) and average postoperative pain at rest (β = 0.18; 95% CI: 0.01 to 0.34) had a positive impact on the postoperative anxiety in patients. However, the patients’ age was negatively correlated with postoperative anxiety (β = −0.19; 95% CI: −0.35 to −0.02). Three variables were found to predict preoperative parental anxiety: their trait anxiety (β = 0.41; 95% CI: 0.23 to 0.59), female gender (β = 0.18; 95% CI: 0.002 to 0.36) and the intravenous route for the postoperative pain management in the patients (β = −0.18; 95% CI: −0.36 to −0.001). The parental postoperative anxiety was influenced by their trait anxiety (β = 0.24; 95% CI: 0.04 to 0.43), preoperative anxiety in patients (β = 0.21; 95% CI: 0.02 to 0.40) and female gender of children (β = 0.19; 95% CI: 0.001 to 0.39). Conclusions. Trait anxiety was a strong factor positively affecting the perioperative state anxiety. In addition, paediatric patient anxiety before surgery was related to their parents’ anxiety, and, after surgery, this was associated with high preoperative anxiety, pain and age. The parents’ anxiety before surgery was influenced by gender and the type of postoperative analgesia in the patients, while, after surgery, this was influenced by the patients’ preoperative anxiety/gender.

## 1. Introduction

Pectus excavatum, also known as funnel chest, and pectus carinatum, or pigeon chest, are the most commonly seen congenital chest wall deformities, with a 5 to 1 and 4 to 1 predominance in males, respectively. Pectus excavatum accounts for approximately 90% of all congenital deformities of the chest wall and is characterised by depression of the sternum and lower costal cartilages, while pectus carinatum is a condition in which the sternum protrudes anteriorly [1]. A mixed-form deformity occurs sporadically. The above-mentioned malformations may present at any time between birth and early adolescence [2].

Paediatric patients with mild, moderate or severe deformities of the thoracic wall pectus have an obvious impairment in their appearance compared to healthy peers. They usually have lower body weight and poorer posture [3]. Self-awareness of deformities and the feeling of being ‘’different’’ make patients avoid exposing their chests in public places [4], which translates into low physical activity [3] and causes psychosocial (anxious-depressed) and social problems [4].

Surgery is still the most effective option for the treatment of patients with chest deformities. Consistent with the literature, cosmetic reasons are the most frequent motivational factors for patients to undergo surgical correction of a chest wall deformity [5]. Surgical repair can be based on both minimal access surgical techniques [6] and open surgery, such as the Ravitch procedure [7,8] and its modification, e.g., by Buchwald [2]. The period before and after surgery may be a very traumatic event for paediatric patients and their parents, who may therefore develop symptoms of anxiety.

According to Spielberger et al., there are two complementary concepts of anxiety: state anxiety and trait anxiety. State anxiety has been described as a temporary reaction to threatening situations (e.g., surgery), consisting of feelings of apprehension, tension, worry and increased activity of the autonomic nervous system. In turn, trait anxiety is understood as a stable personality feature, i.e., the stable tendency to manifest the above-mentioned negative emotions across many situations. It is likely that trait anxiety modulates state anxiety [9]. There is evidence that these two types of anxiety are mapped differently in the human brain [10].

Several studies have examined which sociodemographic and clinical factors are correlated with perioperative anxiety in children and adolescents undergoing elective surgery, as well as in their parents. Liu et al. identified a younger age, the level of parental anxiety, negative previous hospitalisations, less sociableness and the surgical setting as the risk factors which had a significant impact on the children’s anxiety before surgery [11]. On the other hand, the results of the investigation conducted by Caumo et al. revealed that high levels of preoperative anxiety in children and the intensity of postoperative pain were the risk factors for anxiety during the postoperative period [12]. High perioperative anxiety in children may lead to numerous adverse clinical, psychological and behavioural effects during hospitalisation (delirium, worsening of postoperative pain, greater need for analgesia), as well as after discharge from the hospital (separation anxiety, sleep disturbances, aggression, nocturnal enuresis) [13,14,15].

In accordance with the literature, parents’ preoperative anxiety tends to be associated with the female gender, child’s age and other such factors as fear of postoperative pain in children and how to manage it, information about surgery/anaesthesia and information about the possible complications that may occur during and after the child’s surgery [16,17]. Different studies report that parental anxiety levels correlate with children’s postoperative pain [18,19] and can be alleviated by fulfilling care needs [20].

The factors related to preoperative anxiety in paediatric patients undergoing the modified Ravitch procedure and perioperative parental anxiety are unknown. There is very limited data on risk factors related to anxiety after surgical correction of a chest deformity in children [21]. A better understanding of the role of perioperative anxiety risk factors is necessary to identify those factors that may be modifiable. Therefore, the aim of the study was to investigate the anxiety factors in paediatric patients and their parents before and after modified Ravitch surgery and the factors related to the satisfaction of patients with the postoperative care.

## 2. Methods

### 2.1. Trial Design, Setting, Participants

This study is part of a double-blinded, randomised controlled trial to investigate the effects of gabapentin on postoperative outcomes (Clinicaltrails.gov ID: NCT03393702). The protocol of the study was approved by the local ethics committee (decision numbers: KB-6/2017; KB-125/2019).

Patients who were scheduled for a surgical correction of an anterior chest deformity using the modified Ravitch method were recruited between May 2017 and December 2020 by the Department of Thoracic Surgery of the Institute of Tuberculosis and Lung Disease, Rabka Zdrój Branch. The children, 9 to 17 years of age, and their parents (mothers or fathers) were informed about the possible risks and benefits of the study and that participation is voluntary and refusal to participate is possible at any time and is not associated with any penalties. This information was provided both orally and in writing. Before signing the informed consent form (parents, patients age ≥16), it was ensured that the study participants fully understood what was proposed. The study included patients with postoperative chest drainage, in whom intraoperative and postoperative analgesia was provided either by the epidural or intravenous route. The exclusion criteria for participation in this study were known allergies to any of the study medications, a history of chronic pain or daily analgesic use, a diagnosis of psychiatric disorders or epilepsy, those treated oncologically, with impaired verbal communication or those with an American Society of Anesthesiologists’ physical status > 3.

### 2.2. Modified Ravitch Method According to Buchwald

The Buchwald modification [2] of the original procedure described by Ravitch [7,8] is an open technique utilised for the correction of both pectus excavatum (Photograph 1) and pectus carinatum, or their mixed form. The main highlights of this procedure are: an oblique surgical incision in the midline of the sternum (Photograph 2), dissection of the pectoral muscles to expose the sternum and ribs (Photograph 3), cutting off the xiphoid process from the body of the sternum, the resection of the deformed costal cartilages, cutting on both sides of the lateral edges of the sternum wedge-shaped recesses (Photograph 4), giving the peripheral ends of the costal cartilages the shape of the blade (Photograph 5), suture fixation of the costal cartilages to the sternum, and suture fixation of the xiphoid process to the body of the sternum (Photograph 6), and suture fixation of the pectoral muscle to the sternum. A drain was left in the lower pole of the wound in order to evacuate the serous-blood fluid from the retrosternal space and/or air in case of opening of the pleural cavity (Photograph 7).

### 2.3. Postoperative Analgesia

The multimodal, standardised protocol for preoperative, intraoperative and postoperative drug use was detailed in our previous article [22]. Postoperative analgesia was provided by the intravenous or epidural route. An intravenous morphine infusion was administered with a dose of 0.02–0.06 mg/kg per hour. An epidural infusion of 0.2% ropivacaine and fentanyl 5 μg/mL was given with a flow rate of 0.1 mL/kg per hour. In addition, 50% of the patients in each subgroup received perioperative gabapentin.

### 2.4. Variables

Five dependent variables were selected in this study: the preoperative state anxiety of the paediatric patients, the preoperative state anxiety of the parents, the postoperative state anxiety of the paediatric patients, the postoperative state anxiety of the parents, the patient’s satisfaction. The independent variables included both demographic (age, gender) and clinical variables such as: trait anxiety, Body-Mass Index (BMI), American Society of Anesthesiologists’ physical status (ASA), the duration of the anaesthesia/surgery, the type of analgesia (intravenous, epidural), perioperative gabapentin use, postoperative pain intensity, haemodynamic parameters and the number of adverse events of analgesia.

The level of anxiety was measured using the Polish adaptation of the questionnaires: the State-Trait Anxiety Inventory questionnaire (STAI) [23] and the State-Trait Anxiety Inventory for Children (STAI-C) [24]. The first questionnaire is intended for parents and patients 15 years of age and over, and the second is for patients 9 to 14 years of age. Each of these self-assessment questionnaires consists of two 20-item subscales, one of which measures anxiety as a trait, and the other one measures anxiety as a state. All the items in the STAI are rated on a 4-point scale, while, in the STAI-C, they are measured by 1, 2 or 3 points, with a higher score indicating stronger symptoms of anxiety. The results were analysed by a hospital psychologist. The raw scores (STAI: 20–80 points; STAI-C: 20–60 points) were converted to a sten scale from 1 to 10. Sten scores of 1–4 are considered to indicate low anxiety, 5 and 6 moderate, and sten scores of 7 or more are interpreted as high anxiety. Paper versions of the questionnaires were administered individually before the surgery (trait and state anxiety) and on the third postoperative day (state anxiety), and they took approximately 15–20 min for most of the subjects to complete. The psychometric properties of the Polish version of the STAI and STAI-C, expressed as internal consistency (Cronbach’s alpha) and stability (Pearson’s r correlation coefficient), were satisfactory. The Cronbach’s alpha coefficients in the general sample of adults (21–69 years of age), adolescents (between the ages of 15–16) and children (between the age of 9–12) exceeded 0.8 for both state and trait anxiety. The Pearson’s r correlation coefficients for state and trait anxiety in adults varied from 0.39 to 0.68 and 0.59 to 0.86, respectively. The values of the coefficient in children for state anxiety were 0.41–0.58 and 0.55–0.84 for trait anxiety.

The level of patient satisfaction with the postoperative analgesia was measured according to the NRS scale, from 0 (completely dissatisfied) to 10 (completely satisfied), four times during the observational postoperative period, which included the day of surgery, as well as the first, second and third postoperative days. The average satisfaction score was calculated. Higher scores depicted higher satisfaction.

Pain was assessed in dynamic conditions using the Numeric Rating Scale (NRS). The patients were asked to choose an integer from 0 to 10 (0 = no pain; 10 = worst pain imaginable) which best reflects the severity of their pain at rest, during deep breathing and when coughing. The frequency of the pain scores was as follows: after surgery for the first 4 h, every hour, then at least every 4 h; on the second and third postoperative days, at least four times a day, 30 min after analgesic administration. The average pain score was calculated for each patient during the postoperative period (postoperative days 0–3). The treatment threshold/cut-off point for moderate pain treatment was arbitrarily set at NRS ≥ 3.

### 2.5. Statistical Analysis and Sample Size

The intergroup differences were evaluated using the chi-square test for the categorical data and the Mann–Whitney and Wilcoxon probability tests for the continuous data. The Shapiro–Wilk test was used to determine the normality distribution of the data. The results for the categorical data were presented as absolute numbers and percentages, while, for the continuous data, as medians and upper and lower quartiles.

The linear relationship between two variables was verified using the Spearman correlation coefficient (rho). The correlation coefficient was interpreted as negligible (<0.1), weak (0.1–0.39), moderate (0.4–0.69), strong (0.7–0.89) and very strong (0.9–1.0) [25].

Multivariable linear regression models were calculated to find the relationships between the pre- and postoperative state anxiety of the paediatric patients/parents and the independent variables, both demographic (age, gender) and clinical. The independent variables with a *p*-value ≤ 0.1 in simple linear regression models were selected and introduced into the forward stepwise regression (the equal probability value for entry and removal was 0.05). The assumptions for calculating multiple regression were met: there was a linear relationship between the dependent variable and each of the independent variables; the data did not show multicollinearity (Variance Inflation Factor < 1.5); the variance of the residuals was constant (White test *p* > 0.05); and the residuals were normally distributed (Shapiro–Wilk test *p* > 0.05) [26]. The minimum required sample size for a multiple regression study is 84, given a desired probability level of 0.05, with a maximum of four predictors in the model, a medium anticipated effect size of 0.15 and a desired statistical power level of 0.8 [27]. The results of all the multivariable regression models were presented as standardised regression coefficients (ß) and their 95% confidence intervals (CI), partial R^2^.

All the calculations were performed using STATISTICA v.13.3 (TIBCO Software Inc. (2017), Kraków, Poland). For all the statistical analyses, values of *p* < 0.05 were considered significant.

## 3. Results

### 3.1. Demographic and Clinical Characteristics of the Study Participants

A total of 96 children/adolescents and their parents took part in this study (Figure 1). The demographic and clinical characteristics of the study participants are shown in Table 1. The median age of the paediatric patients and parents was 14 and 42 years, respectively. Most of the patients undergoing surgery were male (87.5%), while, in the group of parents, the majority were mothers (78.1%). Among the patients undergoing anaesthesia and surgery, the majority (82.3%) were under ASA I and received Ravitch-type repairs for pectus excavatum (85.4%). The median duration of surgery was 130 min, whereas the median duration of anaesthesia was 195 min. The median duration of continuous analgesia infusion was similar in the intravenous and epidural subgroups and ranged from 46 h to 150 h (median 80 [72; 90] hours). In both subgroups, 50% of the patients were administered gabapentin in the perioperative period. The maximum pain was highest on the day of surgery: at rest was 8 (*n* = 1); during deep breathing, the highest pain value was 6 (*n* = 1); and, when coughing, it was 5 (*n* = 1). The average pain during the postoperative period, both at rest and under dynamic conditions, was below 1/10. The median of the number of adverse events of analgesia was 1 [1; 2]. The adverse events included nausea and vomiting (*n* = 56), urinary retention requiring pharmacological provocation or bladder catheterisation (*n* = 29), fever (*n* = 14), bradycardia (*n* = 6), pruritus (*n* = 5), dizziness (*n* = 2) and harmless neurological complications in the patients treated via the epidural route (*n* = 12).

### 3.2. Preoperative State Anxiety

The distribution of the state anxiety scores in the participants is shown in Figure 2. The prevalence of low, moderate and high levels of anxiety among the children and adolescents before the modified Ravitch procedure was 11.4% (*n* = 11), 43.7% (*n* = 42) and 44.8% (*n* = 43), respectively. In the parents, moderate and high levels of preoperative anxiety were reported in 41 (42.1%) and 42 (43.7%) cases, respectively. Higher preoperative anxiety scores were presented by mothers in comparison to fathers (median 6 [6; 8] vs. 6 [4; 7]; Z = 2.48; *p* = 0.013), the parents of girls compared to boys (median 8 [6; 9.5] vs. 6 [5; 7]; Z = −2.42; *p* = 0.015) and parents of the patients qualified for epidural rather than intravenous analgesia (median 7 [6; 8] vs. 6 [5; 7]; Z = −2.79; *p* = 0.005).

The analysis demonstrates a statistically significant positive correlation between the trait and state anxiety of the patients (rho = 0.49; t = 5.45; *p* < 0.001) and parents (rho = 0.36; t = 3.76; *p* = 0.0003). In the first case, the strength of the correlation was moderate, and in the second, weak. Lastly, there was a weak positive correlation between the parental and children state anxiety (rho = 0.31; t = 3.14; *p* = 0.002) and a negative correlation between the anxiety of the parents and their age (rho = −0.29; t = −2.91; *p* = 0.004).

#### 3.2.1. Factors Related to the Preoperative State Anxiety of Paediatric Patients

The multivariable linear regression model for the state anxiety in the paediatric patients before surgery (Table 2) included the trait anxiety of the patients and the preoperative state anxiety of their parents. This model explained 27% of the variance in the patients’ preoperative anxiety. There was no association between the patients’ preoperative anxiety and their age, ASA and gender (*p* > 0.05).

#### 3.2.2. Factors Related to the Preoperative State Anxiety of Parents

Based on the multiple linear regression analysis (Table 3), three variables predicting the preoperative state anxiety of the parents were found: trait anxiety and female gender of parents (positive regression coefficient), as well as the intravenous route for postoperative pain management in the paediatric patients (negative regression coefficient). The adjusted coefficient of determination indicates that 28% of the measured variance can be explained by these independent variables.

### 3.3. Postoperative State Anxiety

The distribution of the anxiety scores in the paediatric patients and their parents after surgery on postoperative day 3 is presented in Figure 3. Low, moderate and high levels of anxiety were diagnosed in 28 (29.2%), 50 (52.1%) and 18 (18.7%) patients, respectively. The majority of the parents suffered from moderate anxiety (*n* = 45; 46.9%) after the surgery, whereas 1 in 8 parents had a high level of anxiety (*n* = 12; 12.5%).

The level of state anxiety on postoperative day 3 was significantly lower than before surgery in both the patients (median 6 [4; 6] vs. 6 [5; 7]; Z = 5.45; *p* < 0.001) and parents (5 [3; 6] vs. 6 [5.5; 8]; Z = 6.11; *p* < 0.0001).

The results of the correlation studies show that a moderate positive correlation exists between the trait and postoperative state anxiety scores of the patients (rho = 0.43; t = 4.64; *p* < 0.001), whereas, in the parents, the correlation between these variables was weak (rho = 0.32; t = 3.25; *p* = 0.001). There were significant weak positive correlations between the state anxiety scores of the parents and the pre- (rho = 0.30; t = 3.01; *p* = 0.003) and postoperative anxiety scores of the children (rho = 0.26; t = 2.64; *p* = 0.009). The median anxiety scores of the boys’ parents were lower compared to the girls’ parents (5 [3; 6] vs. 6 [5.5; 7.5]; Z = −2.61; *p* = 0.009).

The results of this study reveal that there are no significant correlations between the levels of postoperative state anxiety of the patients and the values of their haemodynamic parameters (heart rate, systolic and diastolic blood pressure) (*p* > 0.05).

#### 3.3.1. Factors Related to the Postoperative State Anxiety of Paediatric Patients

The multiple linear regression model for the state anxiety of the children and adolescents after surgery (Table 4) is statistically significant (*p* < 0.001) and explains 36% of the variance in the patients’ anxiety. The average postoperative pain at rest, trait anxiety of the patients and high level of preoperative state anxiety (vs. low and moderate) have a positive impact on postoperative anxiety. However, the patients’ age negatively correlates with anxiety.

#### 3.3.2. Factors Related to the Postoperative State Anxiety of Parents

Taking into account the findings of the multiple linear regression (Table 5), it was found that the trait anxiety of the parents, preoperative state anxiety of the patients and female gender of the patients significantly predicted the parental postoperative anxiety. However, it should be emphasised that the hereby proposed regression model explained no more than 16% of the variance in parental anxiety. The median age of the girls was 13.5 [12; 15] years, and their mothers were their guardians.

### 3.4. Postoperative Patient Satisfaction

The median of the average patient satisfaction scores during postoperative days 0–3 was 9.7 [8.8; 10]. The satisfaction scores negatively correlated with the trait and state anxiety of the patients and the intensity of pain measured under dynamic conditions (during breathing and when coughing). The strength of the relationship between the variables was weak: rho ranged from −0.21 to −0.27 (Table 6).

## 4. Discussion

The current study identified factors related to anxiety before and after the modified Ravitch procedure in paediatric patients and their parents. Trait anxiety was a strong factor positively affecting both pre- and postoperative state anxiety. In addition, parental anxiety, the level of children’s preoperative anxiety, postoperative pain and younger age were identified as risk factors for perioperative anxiety in the paediatric patients. The perioperative anxiety of the parents depended on the gender, type of children’s analgesia and children’s anxiety before surgery.

Consistent with the results of the current study, Charana et al. [28] identified that high trait anxiety is a possible risk factor for preoperative anxiety in Greek parents whose children were undergoing minor surgery. Scrimin et al. [29] reported that state anxiety in Italian parents of children who had undergone minor/major surgery in the previous 24 h was also predicted by trait anxiety. It is worth noting, based on the results of this study, that higher trait anxiety is associated with higher perioperative anxiety, not only in parents but also in paediatric patients; these findings confirm the hypothesis that trait anxiety and state anxiety are a unidimensional construct [30]. However, some researchers did not observe a relationship between children’s preoperative trait anxiety and postoperative anxiety in the early postoperative period [12]. The reasons for individual differences in the level of trait anxiety are not fully understood. It is assumed that the experience of anxiety depends extensively on cognitive processes, but the involvement of a genetic factor is not excluded [31].

The result of this study shows that parental anxiety is the independent risk factor for preoperative anxiety in children. This is in line with the findings of recent studies [28,32]. High STAI state anxiety scores of parents were found to be a significant risk factor for increased anxiety in the Greek population of children 1–14 years of age before such surgeries as phimosis, hypospadias, inguinal hernia, hydrocele and undescended testicle repair. Getahun et al. [32] found that Ethiopian children 2–12 years of age, whose parents were anxious, had more than three times higher incidence of anxiety in the operating room before the introduction of anaesthesia for general, ophthalmic, urological and other surgeries compared to those whose parents were not anxious. Thus, parental anxiety is an important target for medical staff intervention to alleviate a child’s anxiety [15]. However, it is worth noting that the level of preoperative anxiety in our children was variable when predicting parental anxiety in the postoperative period.

According to the findings of the present study, a high level of preoperative state anxiety in patients and pain intensity measured at rest after surgery were factors clearly related to higher anxiety levels during the postoperative period. This is in agreement with the results of the study by Caumo et al. [12], who reported that Brazilian patients 7–13 years of age, who were more anxious before a variety of elective surgical procedures, showed a three-fold higher risk of postoperative anxiety. The risk of anxiety was approximately 14-fold higher when children suffered from moderate to severe postoperative pain. Both perioperative anxiety and postoperative pain modulated our patients’ satisfaction with postoperative analgesia. This was also observed in a study on adults [33].

Age is believed to be negatively correlated with preoperative anxiety, which means that anxiety is higher in younger patients and decreases with increasing age. Ahmadipour et al. [34] reported that the incidence of anxiety in children 9–12 years of age was approximately three times higher than those 12–18 years of age. The findings of a study by Cui et al. support this observation: preschool children were more anxious than school-age children [35]. The present study failed to find a relationship between age and preoperative state anxiety scores, which is contradictory to the results of the above-mentioned authors. The current study showed—similar to a recent study conducted in children and adolescents after thoracic surgery—that a negative correlation between age and postoperative anxiety exists [21]. Younger children are more susceptible to developing anxiety because of their poorer cognitive and communication abilities, greater reliance on other people, no self-control, little life experience and incomprehension of the principles of the health care system [11].

Gender, in accordance with the literature and this study, is a factor related to preoperative anxiety in parents [17,29] but not in children [28,36]. We noticed that the female parents had significantly higher anxiety scores than male parents. This is consistent with the findings demonstrated by Pomicino et al. [17]: mothers had a four-times greater chance of getting higher results on the anxiety scale than their partners. Moreover, it seems that the gender of the children also influences the anxiety in parents during the postoperative period: the mothers of girls were more anxious than the parents of boys. Similar gender-reliant patterns, but during the preoperative period, were observed by Charana et al. [28] and Erkılıc et al. [37]. Women’s greater vulnerability to anxiety can be partly rationalised by genetic factors, hormonal influences and sociocultural influences [38]. Analysing the results from the socio-cultural perspective, we can assume that the higher level of anxiety among mothers is a consequence of their greater burden resulting from performing traditional female roles (as primary caregivers staying with their child in hospital) and beliefs about the construct of femininity (focus on the body, sexuality and normative standards of beauty). Concern about the appearance of the chest (the size of the scar, the shape of the chest) was probably the reason for the greater fear observed in mothers, especially in the context of the stereotypical perception of the role of a woman in society.

In the presented study, preoperative anxiety was higher in the group of parents who had given informed consent for epidural analgesia compared to those whose children had pain relief via the intravenous route. The reason for the greater anxiety of parents was probably the fear of the risk of serious neurological complications with a permanent adverse outcome [39]. It should be noted that, on the third postoperative day, the type of analgesia had no effect on the level of parents’ anxiety. This can be explained by the fact that the parents who accompanied their children all the time could only observe, in isolated cases, benign neurological complications, such as paraesthesia, anisocoria or Horner’s syndrome, which resolved spontaneously. Prior to the surgery, each parent was informed, among other things, about post-operative pain management options and the potential complications of analgesia verbally by the anaesthesiologist and the anaesthesiology nurse, as well as in writing. They also had the opportunity to familiarise themselves with the epidural catheter. However, individual needs for information are difficult to quantify; the researchers found that insufficient information about anaesthesia was associated with higher anxiety [16].

### Strength and Limitations

To our knowledge, this study is the first research assessing the risk factors related to perioperative anxiety in the parents of children undergoing the Ravitch procedure. It complements the knowledge about the problems encountered in perioperative care in this group of patients [22]. The fact that only one parent participated in the research is a limitation of the study. In addition, the level of anxiety in parents and children could be influenced by factors other than the surgical procedure (e.g., family situation and the scope of information provided on perioperative care).

## 5. Conclusions

Trait anxiety is a strong factor affecting perioperative state anxiety in paediatric patients and their parents. The patient’s anxiety before surgery was also related to the parents’ anxiety, while, after surgery, this was associated with a high level of preoperative anxiety, the intensity of pain and age. The parents’ preoperative anxiety was influenced by gender and the type of postoperative analgesia in the children, while, after surgery, by the children’s preoperative anxiety and children’s gender. More research is needed to confirm these findings and identify other potential determinants of anxiety in children undergoing the Ravitch procedure and their parents.

## 6. Relevance to Clinical Practice

Children undergoing the modified Ravitch procedure, and their parents, deserve special attention as a group exposed to high levels of anxiety. The routine assessment of perioperative anxiety should be a part of good clinical practice. STAI and STAIC are useful screening tools to identify children and parents with high levels of trait anxiety and state anxiety. People with high levels of trait anxiety usually experience greater pre- and post-operative anxiety due to their tendency to interpret a wider range of situations as dangerous or threatening. The identification of these people is important, as is the knowledge of other risk factors (female sex, child’s age, type of analgesia, intensity of postoperative pain) in order to implement psychological interventions tailored to the individual needs of the children and their parents. This will allow them to strengthen their coping mechanisms before surgery and facilitate recovery after surgery. The children and parents should also have an opportunity to consult a psychologist at every stage of treatment.

Nurses and physicians should also pay attention to the way in which they communicate with parents who expect comprehensive, clear, and easy-to-understand information about perioperative care [40]. Providing parents with detailed information in an understandable way can reduce their level of anxiety [41], especially when invasive methods of pain relief are used. It is important to remember that communication is done not only with words but also with body language. The intense nature of contact with patients can cause emotional exhaustion [42] and burnout [43] in healthcare professionals, which, in turn, can be a barrier to effective communication. Therefore, in order to ensure a high quality of patient care, nurses and physicians should receive instrumental support from the organisation (e.g., by providing training on effective communication with the patient and how to deal with stressful situations).

## Figures and Tables

**Figure 1 ijerph-19-16701-f001:**
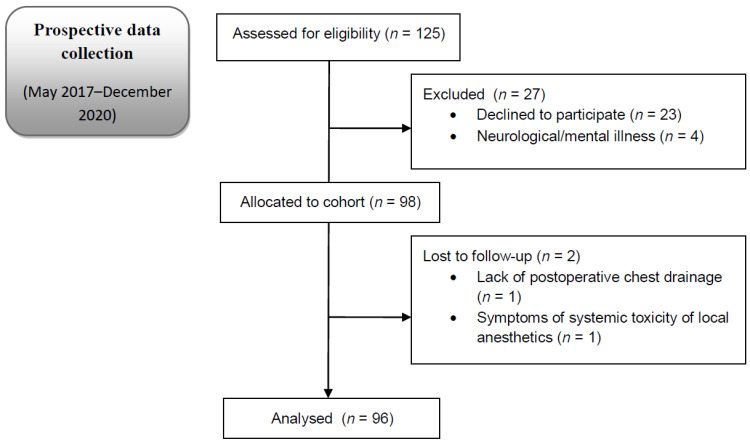
Flow diagram.

**Figure 2 ijerph-19-16701-f002:**
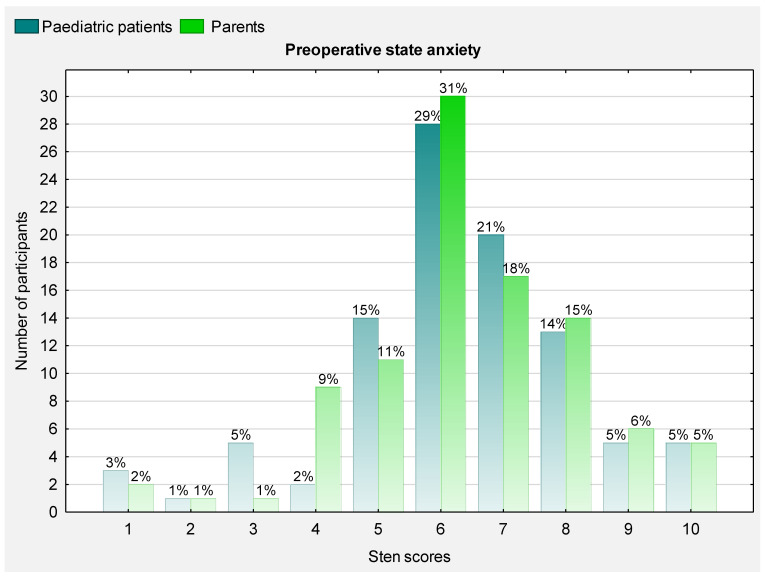
Histogram of the distribution of state anxiety scores in paediatric patients and parents before surgery (1–4 = low level of anxiety; 5–6 = moderate level of anxiety; ≥7 high level of anxiety).

**Figure 3 ijerph-19-16701-f003:**
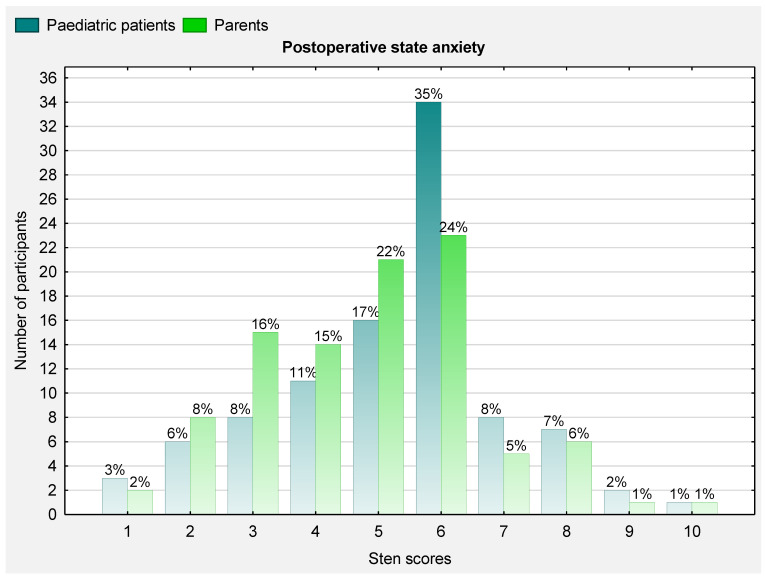
Histogram of the distribution of state anxiety scores in paediatric patients and parents after surgery (1–4 = low level of anxiety; 5–6 = moderate level of anxiety; ≥7 high level of anxiety).

**Table 1 ijerph-19-16701-t001:** Demographic and clinical characteristics of the study participants.

Variables	Paediatric Patients *n* = 96	Parents *n* = 96
Age (years)	14 [13; 16]	42 [39; 46]
Gender (%)		
Female	12 (12.5)	75 (78.1)
Male	84 (87.5)	21 (21.9)
Trait anxiety (sten; range 1–10)	5 [4; 6]	4 [3; 5]
Trait anxiety ≥ 7 sten	19 (19.8)	8 (8.3)
Preoperative state anxiety (sten; range 1–10)	6 [5; 7]	6 [5.5; 8]
Postoperative state anxiety (sten; range 1–10)	6 [4; 6]	5 [3; 6]
Type of pectus deformities		
Pectus excavatum	82 (85.4)	
Pectus carinatum	12 (12.5)	
Mixed	2 (2.1)	
American Society of Anesthesiologists’ physical status (ASA; %)		
ASA 1	79 (82.3)	
ASA 2	17 (17.7)	
Body-Mass Index (BMI; kg/m^2^)	18 (16.3; 19.6)	
Perioperative gabapentin		
Intravenous subgroup	28 (50%)	
Epidural subgroup	20 (50%)	
Duration of anaesthesia (min)	195 [175; 210]	
Duration of surgery (min)	130 [115; 150]	
Postoperative analgesia		
Intravenous route	56 (58.3)	
Epidural route	40 (41.7)	
Average postoperative pain (NRS, range 0–10)		
At rest	0.3 [0.1; 0.6]	
During deep breathing	0.2 [0.1; 0.3]	
When coughing	0.3 [0.1; 0.5]	
Postoperative average heart rate [beats/min]	77 [70; 84]	
Postoperative average systolic blood pressure [mmHg]	109 [105; 114]	
Postoperative average diastolic blood pressure [mmHg]	61 [58; 64]	
Number of adverse events of analgesia	1 [1; 2]	

Results presented as absolute numbers (percentages) or median [upper and lower quartile].

**Table 2 ijerph-19-16701-t002:** Multiple linear regression analysis for variables predicting the preoperative state anxiety of paediatric patients.

Factors	Simple Regression ß (95% CI)	Multiple Regression ß (95% Cl)	Partial R^2^	Model
Trait anxiety of patients	0.48 (0.30 to 0.65) **	0.47 (0.29 to 0.64) **	0.23	R^2^ = 0.27, F (2,93) = 18.78; *p* < 0.0001
Preoperative state anxiety of parents	0.26 (0.06 to 0.46) *	0.24 (0.07 to 0.42) *	0.08	
ASA 1 ^reference category: ASA 2^	−0.04 (−0.24 to 0.16)			
Female gender	0.12 (−0.08 to 0.32)			
Patients’ age	−0.12 (−0.33 to 0.08)			

ß—standardised regression coefficient; Cl—confidence interval; R^2^—adjusted coefficient of determination; *p* < 0.05 *; *p* < 0.001 **.

**Table 3 ijerph-19-16701-t003:** Multiple linear regression analysis for variables predicting the preoperative state anxiety of parents.

Factors	Simple Regression ß (95% CI)	Multiple Regression ß (95% Cl)	Partial R^2^	Model
Trait anxiety of parents	0.47 (0.29 to 0.66) **	0.41 (0.23 to 0.59) **	0.17	R^2^ = 0.28; F (3,91) = 13.00; *p* < 0.0001
Female gender of parents	0.31 (0.11 to 0.51) *	0.18 (0.002 to 0.36) *	0.04	
Intravenous analgesia of patients ^reference category: epidural analgesia^	−0.25 (−0.45 to−0.05) *	−0.18 (−0.36 to −0.01) *	0.04	
Female gender of patients	0.27 (0.07 to 0.47) *			
Preoperative state anxiety of patients	0.24 (0.04 to 0.44) *			
ASA 1 ^reference category: ASA 2^	−0.20 (−0.40 to 0.004)			

ß—standardised regression coefficient; Cl—confidence interval; R^2^—adjusted coefficient of determination; *p* < 0.05 *; *p* < 0.001 **.

**Table 4 ijerph-19-16701-t004:** Multiple linear regression analysis for variables predicting the postoperative state anxiety of patients.

Factors	Simple Regression ß (95% CI)	Multiple Regression ß (95% Cl)	Partial R^2^	Model
Average pain at rest (postoperative days 0–3)	0.23 (0.03 to 0.43) *	0.18 (0.01 to 0.34) *	0.05	R^2^ = 0.36; F (4,91) = 14.15; *p* < 0.001
Patient’s age	−0.20 (−0.40 to −0.003) *	−0.19 (−0.35 to −0.02) *	0.05	
Trait anxiety of patients	0.42 (0.24 to 0.61) **	0.22 (0.04 to 0.40) *	0.06	
High preoperative state anxiety of patients ^reference category: low and moderate level^	0.52 (0.35 to 0.70) **	0.40 (0.22 to 0.58) **	0.17	
Postoperative state anxiety of parents	0.23 (0.03 to 0.43) *			
Intravenous analgesia ^reference category: epidural analgesia^	−0.19 (−0.39 to 0.008)			

ß—standardised regression coefficient; Cl—confidence interval; R^2^—adjusted coefficient of determination; *p* < 0.05 *; *p* < 0.001 **.

**Table 5 ijerph-19-16701-t005:** Multiple linear regression analysis for variables predicting postoperative parental anxiety.

Factors	Simple regression ß (95% CI)	Multiple regression ß (95% Cl)	Partial R^2^	Model
Trait anxiety of parents	0.32 (0.12 to 0.51) **	0.24 (0.04 to 0.43) *	0.06	R^2^ = 0.16, F (3, 92) = 7.12; *p* < 0.0001
Preoperative state anxiety of patients	0.27 (0.08 to 0.47) **	0.21 (0.02 to 0.40) *	0.05	
Female gender of patients	0.28 (0.08 to 0.47) **	0.19 (0.001 to 0.39) *	0.04	
Postoperative state anxiety of patients	0.23 (0.03 to 0.43) *			
Preoperative state anxiety of parents	0.30 (0.11 to 0.50) **			
Trait anxiety of patients	0.25 (0.05 to 0.45)*			

ß—standardised regression coefficient; Cl—confidence interval; *p* < 0.05 *; *p* < 0.001 **; R^2^—adjusted coefficient of determination.

**Table 6 ijerph-19-16701-t006:** Spearman’s rank correlations for analysis of the association between patient satisfaction from postoperative analgesia and anxiety and postoperative pain.

Variables	rho	t	*p* Value
Trait anxiety of patents	−0.27	−2.71	0.008
Preoperative state anxiety of patients	−0.21	−2.04	0.043
Postoperative state anxiety of patients	−0.26	−2.58	0.011
Average pain during breathing	−0.21	−2.11	0.037
Average pain when coughing	−0.24	−2.41	0.017

rho: Spearman coefficient.

## Data Availability

A dataset will be made available upon request to the corresponding authors one year after the publication of this study. The request must include a statistical analysis plan.

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
