# Peer review of "Factors Related to Anxiety in Paediatric Patients and Their Parents before and after a Modified Ravitch Procedure—A Single-Centre Cohort Study"

_ijerph, 2022, doi:10.3390/ijerph192416701_

Round 1
Reviewer 1 Report
Outstanding rigor and quality of application of multiple regression analysis. Very clearly written and organized. Relevant recommendations for practical application are included. I might also include one or two recommendations for future related research. I highly recommend acceptance for publication.
Author Response
Dear Reviewer,
We would like to thank you for review our manuscript. As suggested, we have added recommendations for future research work:
''More research is needed to confirm these findings and identify other potential determinants of anxiety in children undergoing the Ravitch procedure and their parents''.
Reviewer 2 Report
Dear authors,
Thanks for the wonderful piece of work with a detailed pictorial presentation. It is indeed significant to focus on the preoperative anxiety of parents and patients before treatment. I appreciated the practical implications, however, an extended paragraph is recommended in clinical practices for nurses and physicians. For instance, an exact communication style for both parties to handle anxiety, considering the personality type of parents and patients.
Additionally, this is also a job demand for healthcare professionals and more often they do sympathize with attendants. As a result, these professionals perform emotional labor due to the nature of their work. It is recommended to quote references for handling healthcare workers' psychological well-being in future directions, using the following literature.
Impact of nurses’ emotional labour on job stress and emotional exhaustion amid COVID-19: The role of instrumental support and coaching leadership as moderators (2022)
Examining the role of transformational leadership and mission valence on burnout among hospital staff (2021)
Hope my comments will help you to improve the draft.
Author Response
Dear Reviewer,
We would like to thank you for a detailed review of our manuscript and all of your valuable remarks. We have addressed them all in detail below.
Point 1:
Thanks for the wonderful piece of work with a detailed pictorial presentation. It is indeed significant to focus on the preoperative anxiety of parents and patients before treatment. I appreciated the practical implications, however, an extended paragraph is recommended in clinical practices for nurses and physicians. For instance, an exact communication style for both parties to handle anxiety, considering the personality type of parents and patients.
Additionally, this is also a job demand for healthcare professionals and more often they do sympathize with attendants. As a result, these professionals perform emotional labor due to the nature of their work. It is recommended to quote references for handling healthcare workers' psychological well-being in future directions, using the following literature.
Impact of nurses’ emotional labour on job stress and emotional exhaustion amid COVID-19: The role of instrumental support and coaching leadership as moderators (2022)
Examining the role of transformational leadership and mission valence on burnout among hospital staff (2021)
Hope my comments will help you to improve the draft.
Response 1: We have updated the reference list as per your recommendation. We have also added an additional comment regarding the barriers to effective communication with the patient ‘’It is important to remember that communication is done not only with words, but also with body language. The intense nature of contact with patients can cause emotional exhaustion [42] and burnout [43] in healthcare professionals, which in turn can be barrier to effective communication. Therefore, in order to ensure high quality of patient care, nurses and physicians should receive instrumental support from the organization (e.g. by providing training on effective communication with the patient and how to deal with stressful situations).’’
Reviewer 3 Report
The manuscript discusses the important factors of anxiety of children and their parents before and after modified Ravich operation, and has certain research significance. The following points need to be improved:
1) In Section 2.4, all variables should be summarized in a table for easy presentation and understanding;
2) The methods or tools for establishing regression models should be described in Section 2.5;
3) Although this work has been approved by the Ethics Committee, the possible ethical risks and the corresponding measures taken in the research should be indicated in the manuscript;
4) It is suggested that the authors explain the future direction of this research in Conclusion.
Author Response
Dear Reviewer,
We would like to thank you for a detailed review of our manuscript and your valuable remarks. We have addressed them below and hope that you will find our explanations sufficient.
Point 1:
The manuscript discusses the important factors of anxiety of children and their parents before and after modified Ravich operation, and has certain research significance. The following points need to be improved:
1) In Section 2.4, all variables should be summarized in a table for easy presentation and understanding.
Response: It seems to us that creating another 7th table is not necessary because the variables are clearly described.
2) The methods or tools for establishing regression models should be described in Section 2.5.
Response: This was clearly described in the original manuscript.
3) Although this work has been approved by the Ethics Committee, the possible ethical risks and the corresponding measures taken in the research should be indicated in the manuscript.
Response: We have added in revised Methods (2.1. Trial Design, setting, participants): ‘’Children 9 to 17 years of age and their parents (mothers or fathers) were informed about the possible risks and benefits of the study and that participation is voluntary and refusal to participate is possible at any time, and is not associated with any penalties. This information was provided both orally and in writing. Before signing the informed consent form (parents, patients age ≥16), it was ensured that study participants fully understood what was proposed.’’
4) It is suggested that the authors explain the future direction of this research in Conclusion.
Response: We have added recommendations for further research work: ‘’More research is needed to confirm these findings and identify other potential determinants of anxiety in children undergoing the Ravitch procedure and their parents.’’